# Canonical and Noncanonical ER Stress-Mediated Autophagy Is a Bite the Bullet in View of Cancer Therapy

**DOI:** 10.3390/cells11233773

**Published:** 2022-11-25

**Authors:** Rashedul Alam, Mohammad Fazlul Kabir, Hyung-Ryong Kim, Han-Jung Chae

**Affiliations:** 1Department of Pharmacology, Medical School, Jeonbuk National University, Jeonju 54896, Republic of Korea; 2School of Pharmacy and Institute of New Drug Development, Jeonbuk National University, Jeonju 54896, Republic of Korea; 3Department Biochemistry & Molecular Biology, Hollings Cancer Center, Medical University of South Carolina, Charleston, SC 29425, USA; 4Department of Biological Science, St. John’s University, New York, NY 11439, USA; 5Department of Dental Pharmacology, College of Dentistry, Jeonbuk National University, Jeonju 54896, Republic of Korea

**Keywords:** autophagy, ER stress, UPR, tumerogenesis, cancer therapy

## Abstract

Cancer cells adapt multiple mechanisms to counter intense stress on their way to growth. Tumor microenvironment stress leads to canonical and noncanonical endoplasmic stress (ER) responses, which mediate autophagy and are engaged during proteotoxic challenges to clear unfolded or misfolded proteins and damaged organelles to mitigate stress. In these conditions, autophagy functions as a cytoprotective mechanism in which malignant tumor cells reuse degraded materials to generate energy under adverse growing conditions. However, cellular protection by autophagy is thought to be complicated, contentious, and context-dependent; the stress response to autophagy is suggested to support tumorigenesis and drug resistance, which must be adequately addressed. This review describes significant findings that suggest accelerated autophagy in cancer, a novel obstacle for anticancer therapy, and discusses the UPR components that have been suggested to be untreatable. Thus, addressing the UPR or noncanonical ER stress components is the most effective approach to suppressing cytoprotective autophagy for better and more effective cancer treatment.

## 1. Introduction

Cancer is a complex disorder defined by unpredictable cell division that can acquire an obtrusive phenotype due to genetic abnormalities, epigenetic changes, and environmental variables that alter the expression or function of encoded products. Endoplasmic reticulum (ER) stress and autophagy have been related to cancer for decades. Now, these concepts dominate the scientific discourse. ER stress induces levels of misfolded proteins in the ER lumen, redox imbalance, and the disruption of Ca^2+^ balance in the ER, triggering an evolutionarily conserved cascade called unfolded protein response (UPR). Here, we define the “canonical ER stress response” that restores ER homeostasis by stimulating ER transmembrane proteins such as ATF6 (activating transcription factor 6), IRE1α (inositol-requiring enzyme 1α), and PERK (protein kinase RNA-like ER kinase) regulated by ER chaperone protein GRP78/BiP. The noncanonical ER stress responses are pathways that intersect UPR components responsible for cytoprotection during stress.

The UPR and its components are triggered by physiological and pathological circumstances such as protein glycosylation arising from glucose deprivation, the reduced production of disulfide bonds during hypoxia, DNA damage, and therapeutic stress [1,2]. UPR signal transduction reactions are designed to reinstate protein homeostasis by inhibiting global protein synthesis, which enhances the folding ability to expand the ER size to reduce ER luminal protein burden. ER expansion has been proposed to relay ER stress, ER Ca^2+^, and ER-phagy receptors [3,4]. ER stress and proteasomal degradation capacity exceeding ER lumen might trigger autophagy and play apoptotic and adaptive signaling functions in eukaryotic cells [5,6]. Autophagy serves a crucial function in maintaining cellular homeostasis by eliminating harmful substances such as damaged organelles, protein aggregates, and intracellular pathogens from the cytoplasm for lysosomal degradation in order to adapt to metabolic stress and prevent genetic damage. ER homeostasis and autophagy are crucial for tumor development, metastasis, and chemo-resistance in multiple cancer types [7,8,9,10]. Nevertheless, unresolved ER stress induces programmed cell death (PCD) to prevent the spread of genomic insults. Moreover, the connections between ER stress and cancer are highlighted in several reports where UPR appears to regulate the dynamic microenvironment of tumor growth. Additionally, multiple signals emanate from UPR to induce autophagy, which exerts a cytoprotective effect. Thus, targeting UPR components is a viable approach for the development of novel cancer therapy. In this review, we evaluated the discourses on the canonical and noncanonical ER stress responses to autophagy, the biological functions of autophagy, and its perspective pathophysiology in cancer to exploit potential positive clinical outcomes.

## 2. Molecular Mechanism of Autophagy

Autophagy involves complicated and sequential molecular processes, and understanding these processes is critical for developing potential novel therapeutic applications. Autophagy is associated with multiple cellular and external irregularities such as cargo selection and packaging, phagophore membrane expansion, vesicle elongation, vesicle nucleation, and the fusion of matured autophagosomes with the lysosome during cell homeostasis [11,12]. Autophagy-related genes (ATGs) regulate autophagy processes. Autophagosome production in mammalian cells is induced by amino acid (AA) deprivation or other environmental signals by inhibiting the mammalian target of rapamycin complex 1 (mTORC1) [13]. The mTORC1 is a crucial regulator of macroautophagy that inhibits autophagosome formation by accelerating the inactivation of ATG13 and ULK1 phosphorylation under standard conditions [14,15]. In addition, the stress-induced dephosphorylation of mTORC1 leads to its detachment from the complex. Further, autophosphorylation at the active sites of ULK1/2 activates FIP200 and ATG1 [16,17].

The AMP-activated protein kinase (AMPK) potentially prevents mTORC1 activity whenever the cellular ATP/AMP ratio falls to lower levels [18,19]. A multipronged inhibitory network is disrupted when mTORC1 is inactivated by AMPK, resulting in decreased ATP levels and subsequent AMPK accumulation. Under glucose deprivation-induced ER stress conditions, AMPK catalyzes ULK1 by activating the phosphorylation of Ser 777 and Ser 317 [20]. An active ULK1 complex can also activate the phosphoinositide 3-kinase (PI3K) family, including class I, II, and III. Beclin 1 (BECN1) and vacuolar protein sorting 34 (VPS34) activate AMBRA1 (activating molecule in Beclin 1-regulated autophagy 1), phosphoinositide 3-kinase regulatory subunit 4 (PIK3R4), and ATG14L to facilitate phagosome biogenesis [21]. The activation of VPS34 forms phosphatidylinositol 3-phosphate (PI3P), which helps in the enlargement of autophagosomal membranes till closure by interacting with PI3P-binding ATG and WIPI family proteins [21]. Additional proteins, such as ATG18, ATG2, ATG9, and DFCP1 (double FYVE-containing protein 1), are recruited to the site of phagophore formation as PIP3 concentration rises [22]. The ER acts as a membrane source whenever phagophore synthesis occurs in the presence of DFCP1 and the ATG2-WIPI/ATG18 complex [23]. Then, ATG9 facilitates membrane vesicles from the ER, plasma membrane, and mitochondria [24]. Autophagosome precursor synthesis in the plasma membrane implicates interactions between ATG16L and the heavy chain clathrin. In contrast, ATG5 and LC3 localization to the outer mitochondrial membrane is obligatory for phagophore generation in mitochondria.

Further, pre-autophagosome elongation and maturation are also influenced by LC3/ATG8. ATG4 converts proLC3 into LC3-I, the active form of LC3 [25,26]. In the presence of ATG7 and ATG3, the ATG12 conjugation complex (ATG12-ATG5-ATG16) acts as an essential component in the elongation and maturation of the autophagosomal membrane. Additionally, in conjunction with phosphatidylethanolamine (PE), ATG12 conjugation complex aids in acquiring and transitioning the autophagosomal membrane’s LC3-I to LC3-II [27]. LC3-II is drawn to the autophagosomal membrane due to its lipid moiety, and unlike LC3-1, LC3-II is not readily accessible in the cytoplasm [28]. The conversion of LC3-I to LC3-II results in an elongated autophagosome (a hallmark for autophagy) and then transforms into a mature autophagosome. ATG8-interacting motif (AIM) and LC3-interacting region (LIR) of p62/sequestosome-1 (p62/SQSTM1), nuclear dot protein 52 (NDP52), breast cancer 1 (BRCA1/NBR1), and optineurin enhance cargo selection and selective autophagy by transporting cargos to nucleation sites [29,30,31,32]. Eventually, autophagosome maturation occurs and fuses with a lysosome. ATG proteins are retrieved from the autophagosome outer membrane and SNAREs (soluble NSF attachment protein receptors), resulting in the fusion of epitopes, and a homotypic fusion and protein sorting (HOPS) complex is recruited [33,34]. Specific proteases decompose these autophagosomal substances in the acidic lysosome environment [35]. Autophagy produces byproducts such as sugars, AA, nucleosides, fatty acids, and macromolecules, which are used by the cell for biosynthesis or to fulfill its energy demands under poor growth conditions allowing cell survival.

## 3. ER Stress Assuage by Autophagy

The ER is a membrane-bound fundamental organelle that serves multiple roles in the cell, including cellular metabolism, signal transduction, protein synthesis, and calcium (Ca^2+^) storage [36]. These functions are subject to extensive and dynamic morphological and compositional sculpting. The ER produces a continuous membrane network consisting of sheets, tubules, and matrices that fluctuates in response to functional demands [37]. Still, several unanswered concerns remain with respect to how the ER alters its shape in response to cellular inputs, cell type, cell cycle status, and development.

ER function and morphology are closely related to autophagy during ER stress. During protein processing and folding assembly, errors such as the disruption of ER proteostasis and the accumulation of aberrant proteins occur, which activates adaptation signaling [38]. ER folding capability and size need to adapt rapidly to environmental and developmental conditions or biosynthetic requirements. This adaptation is achievable by inducing degradative pathways and transcriptional/translational mechanisms known as UPR. The UPR increases the size of the ER and the production of ER-resident proteins that control the multiple ER functions. Proteostasis and autophagy are the fundamental degradation mechanisms that restore cell homeostasis [39]. The ER-associated degradation (ERAD) pathway retro-translocates misfolded ER proteins into the cytosol and are degraded by proteasomes [38]. Moreover, organelles, pathogens, protein clumps, and liquid protein condensates are cleared by autophagy [40,41]. Autophagy is regulated by stress signals and involved in the lysosomal hydrolase-mediated breakdown of aberrant proteins and damaged organelles. However, it is less understood how the lysosomal catabolic activities during ER stress sustain stability in ER size and prevent excessive ER expansion, which ensures physiologic ER size after recovery from ER stress. ER expansion is restricted by ER subdomains such as CLIMP-63, kinectin, and p180. Furthermore, these subdomains are presumed to influence the mounting and luminal alignment of ER sheets [4]. Cells alter CLIMP63 protein levels and p180–microtubule-binding for optimal autophagy response during nutrient deprivation conditions and move ER to lysosomes [42]. Moreover, specific ER subdomains such as lunapark (LNPK), atlastin (ATL), and reticulon (RTN) proteins are responsible for selective lysosomal delivery [43,44,45,46,47]. Hydrophobic hairpin domains/RTN domains are considered to induce the bending of ER tubules and sheets through hydrophobic wedging and scaffolding in REEP, FAM134, and RTN subdomains [48,49]. These domains interact with the LC3/GABARAP family to affect and manage ER shape and content change via the lysosome [50,51,52]. Together, these findings indicate that ER stress and autophagy share a common feature in protecting cells by relieving stress. On the contrary, programmed cell death (PCD) begins soon after stress exceeds threshold levels. Compensatory mechanisms can no longer support the ER function due to its significant impairment.

## 4. Significance of Molecular Mechanisms Regulating ER-Phagy in Selection of Therapeutic Targets

Post-translational modifications (PTMs) have significantly broadened the functional scope of proteins, which are crucial for the immune detection of malignancy therapy. Specifically, multiple peptides introduced to T cells through histocompatibility complex are considered post-translationally transformed peptides. The regulation of these transformations may influence the selection of therapeutic targets and respective immune responses. ATP-dependent enzymes associated with activation, binding, and ligation (E1, E2, and E3) represent a three-step enzymatic mechanism that facilitates the covalent binding of ubiquitin-like small-protein via isopeptide bonds within C-terminal diglycine motifs, ε-amino groups, and associated substrates [53]. Alterations in the ubiquitin system have been linked to multiple diseases, including cancer, where mutations in E3 ligases such as Mdm2, pVHL, and BRCA1 are correlated with disease progression, prognosis, and drug resistance [53].

UFMylation is one such ubiquitin-like modification influencing multiple biological functions. UFM1 conjugates with target proteins via UBA 5 (ubiquitin-like modifier-activating enzyme 5), UFC1 (UFM1-conjugating enzyme 1), and UFL1 (UFM1-specific ligase 1) enzymes [54]. Protein UFMylation is conserved as well as reversible in most eukaryotic organisms, with certain exceptions. UFM1-specific proteases (Ufsps) separate UFM1 from its target proteins [55]. Recent investigations have shown that UFMylation modulates ER stress, vesicle movements, cell-autonomous erythroid differentiation, hematopoiesis, β-oxidation of fatty acids, and GPCR (G-protein-coupled receptor) biogenesis [54]. Collectively, PTMs play a significant role in cancer progression and substantially influence the selection of targets. Thus, PTM-assisted strategies are gaining momentum for enhanced cancer immunotherapy. Additionally, the autophagy pathway is recognized as a druggable mechanism. Still, multiple autophagy genes are linked to drug resistance and have been shown to be challenging for drug development. Hence, addressing the UPR or noncanonical ER stress processes may suppress cytoprotective autophagy more effectively, offering optimism for developing effective cancer therapy.

## 5. Autophagic By-Product Effect on Cellular Stress Tolerance

Autophagy is a dynamic catabolic process, and it has been linked with oncogenesis. Moreover, autophagy is essential for adaptation to fluctuating environmental conditions. Considering its significance, autophagy was proposed for tumor cell renovation and homeostasis. Upon breakdown, autophagy produces metabolic substrates that produce energy for the synthesis of new proteins and membranes [56]. However, little is known about how autophagy modulates the characteristics and the abundance of the substrates and how autophagy itself is governed at the cellular level. Nonetheless, autophagy seems to be an essential energy generator in specific conditions as it provides substrates such as AA to the TCA cycle to produce ATP. For instance, metabolome analysis of Ras-expressing cancer cells demonstrates that autophagy is essential for maintaining TCA cycle metabolites [57]. Moreover, autophagy makes sufficient ATP for the survival of cancer cells through the reutilization of free AA and FFA [58]. Hence, autophagy-deficient tumor-derived cell lines fail to degrade macromolecules, resulting in impaired respiration due to an energy crisis. Additionally, insufficient aspartate blocks the TCA cycle, resulting in mitochondrial dysfunction, which restricts cancer progression [59]. Further, ATF4-mediated integrated stress response (ISR) is triggered by the phosphorylation of eukaryotic initiator of factor 2 (eIF2), which assures the availability of protein and glutathione production. This shields cells from oxidative stress and preserves homeostasis contributing to tumorigenesis and metastasis [60]. Previous reports on ROS and oxidative stress linked with ER stress [61,62] might provide hope for a resurrection of antitumor immunity by stimulating cell death, which could form a novel method in cancer therapy. Still, the activation of ISR plays a significant role in adapting ER functions, stress tolerance, and other metabolic consequences. Several investigations have suggested that autophagy-derived AA plays a key role in the synthesis of proteins required for adaptation during starvation. Additionally, AA conjugates of 1, 2-diselenan-4-amine potentially increase protein disulfide isomerase (PDI) activity [63]. In the ER, PDIs form disulfide bonds having multiple thioredoxin-like domains that are essential for preserving the native structures of proteins. Furthermore, these domains facilitate the proper folding of proteins, which ensures ER homeostasis and effectively inhibits misfolded protein aggregation [64]. Moreover, autophagy modulates FFA availability via the degradation of lipid droplets. These lipid droplets are controlled by the nexus between the ER and the mitochondrial membrane numbers and functions known as the mitochondria–ER contacts or MERCs [65,66]. Thus, contact sides control mitochondrial energy and respiratory sources via diverse oxidizable substrates. For example, fatty acid oxidation is an essential catabolic pathway, and autophagy is a critical regulator for cell growth in acute myeloid leukemia [66]. However, how autophagy is upregulated upon stress and confers to cellular stress tolerance is still undiscovered, and it is the susceptibility of cancer cells to autophagy that determines the novel therapeutic approach to treat cancer.

## 6. Canonical Pathway Regulates ER-Stress Response

UPR or canonical ER stress response is triggered when misfolded proteins are exceeded and relayed via ER transmembrane sensors such as IRE1α, PERK, and ATF6. HSPA5/GRP78/Bip, an essential chaperone, is bound to these UPR sensors and prevents downstream signaling [67]. Critical unfolded protein levels decouple sufficient GRP78 from UPR sensors to activate downstream signaling. These signals determine protein synthesis and enhance protein-folding capacity to reinstate ER homeostasis. Moreover, depending on the trigger type and stress intensity, UPR signaling pathways are strongly connected to cell fate, autophagy, inflammatory response, and oxidative stress [68]. These extensive connections make conditions more complex.

IRE1, the most conserved protein and the oldest branch of the UPR sensor, spans the ER membrane. IRE1 comprises three distinct domains: a luminary N-terminal domain, a serine/threonine kinase cytosolic domain, and an endoribonuclease cytosolic (RNase) domain [69]. IRE1α is an extensively investigated protein, is prevalent in most tissues, and is expressed ubiquitously, whereas IRE1β is found only in limited tissues [67]. Intriguingly, IRE1α presents adaptive or death signals through its endoribonuclease activity, unconventional splicing of XBP1 (sXBP1), and regulated IRE1α-dependent decay (RIDD). In contrast, XBP1 mRNA splicing responds to ER stress with a cytoprotective effect [70]. The aggregation of unfolded proteins activates ER stress response through IRE1α signals to a cytosolic kinase-endoribonuclease module, which forms the transcription factor XBP1s by eliminating a 26-nucleotide intron from mRNA [71]. Following translation, the XBP1-spliced event causes a frameshift in the mRNA, making the transcription factor XBP1 active and stable. Subsequently, as part of a transcriptional program to solve the protein misfolding in the ER, the XBP1 transcription factor determines the expressions of lipid biosynthesis linked genes and ER chaperones [72]. In higher eukaryotes, IRE1α activation triggers the downregulated translation of ER-targeted proteins via the direct degradation of ER-localized mRNAs in a process known as RIDD. In addition to decreasing ER load, mammalian RIDD regulates triglyceride (TG) and cholesterol (CHOL) metabolism, apoptotic signaling via DR5, protective autophagy via the biogenesis of lysosome-associated organelles complex 1 (BLOC1S1/BLOS1), and DNA repair via Ruvbl1 [73,74,75,76,77]. Furthermore, ER stress promotes TNF receptor-associated factor 2 (TRAF2) oligomerization via IRE1α, which activates ASK1-MKK4/7-JNK signaling, promotes survival via c-Jun-dependent Adapt78, and activates c-Jun *N*-terminal kinase (JNK) [78]. Inflammatory response by the IRE1α-TRAF2 complex recruits IkappaB (IκB) kinase, which contributes to the breakdown and phosphorylates IκB [79]. Thus, the translocation of NF-κB to the nucleus regulates inflammatory gene transcription. Additionally, phosphorylated JNK triggers the apoptotic signal via multiple pathways. JNK activation induces cell death by activating cytochrome C-mediated apoptotic pathway via B-cell lymphoma family 2 (Bcl-2) proteins [80].

Among the UPR sensors, PERK interacts with BiP/GRP78 ATPase binding domain. PERK phosphorylates the downstream substrate of the eIF2 at serine 51, inhibiting global protein synthesis [81]. It is a crucial tool in the cell’s arsenal against stress elements such as viral infection, misfolded protein accumulation, and starvation [82]. Participation in translation, eIF2 bound by GTP, and Met-tRNA^met^ facilitates the identification of the start codon by establishing a 43S preinitiation complex with a 40S ribosomal subunit and translation initiation factors such as eIF1, eIF1A, and eIF3 [83]. To restore the activated state, inactive eIF2-GDP interactions turned to active eIF2-GTP interaction, which inhibited the guanine nucleotide exchange (GNE) activity of eIF2B. Further, the inhibitory effects of eIF2α phosphorylation on translation will depend on the levels of eIF2α relative to eIF2B [84]. The eIF2 is phosphorylated at Ser51, which causes the production of specific proteins such as the transcription factor ATF4 [85]. ATF4 mediates several adaptive responses and induces C/EBP homologous protein (CHOP), DNA damage-inducible 34 (GADD34), and growth inhibition, redirecting the response toward cell death [86,87]. Following this function, eIF2α dephosphorylation is related to GADD34 at later stages of the stress response. PERK and IRE1α synergistically activate the NF-κB transcription factor, resulting in the generation of pro-inflammatory cytokines. Under ER stress conditions, cytokine secretions contribute to a range of programs, from inflammation transmission to tumor initiation [85,88].

ATF6 is an ER stress sensor and a type II transmembrane protein that interacts with the resident chaperone BiP/GRP78 to position itself in the ER membrane under normal homeostatic conditions [89]. It has also been associated with stress regulators at the *N*-terminal of the cytosolic domain and a C-terminus in the ER lumen [90]. ER stress stimulates ATF6, the dissociation of C-terminus from GRP78, and ATF6’s move from the ER to the Golgi upon unfolded protein accumulation. Additionally, ATF6 is cleaved in the luminal domain of the Golgi by enzyme site-1 protease (S1P) and then by site-2 protease (S2P) and release [89]. Cleaved ATF6-N cytosolic fragment is transported to the nucleus and binds to ATF/cAMP response elements and cis-acting ER stress response elements (ERSE) in the promoter regions of UPR target genes such as ER-residence chaperones, XBP1, CHOP, PDIs, and ERAD components [69]. Together, ATF6 ensures the adaptive nature by enhancing cellular protein folding and processing capacity to maintain protein homeostasis in the ER.

### The Interplay between Canonical ER Stress and Autophagy in Cancer

Cancer cells encounter multiple stress conditions such as hypoxia, oxidative stress, reduced glucose, deficiency in growth factor, lactic acidosis, AA starvation, and reduced protein-folding abilities in the ER during their growth. Collectively, these conditions are challenging for protein processing in the ER, resulting in ER stress triggering UPR. Several investigations have demonstrated high-level activation in UPR branches demonstrated autophagy-mediated defense mechanisms in human hematopoietic and solid tumors [91,92,93,94]. Several studies have indicated that ER stress enhances lysosomes required for autophagy and stimulates membrane-bound LC3-II expression in different malignancies [95]. Pancreatic ductal adenocarcinoma (PDAC) can be put under consideration when PDAC cell lines activate FAK and JNK that promote penetration, and its downregulation inhibits the invasive capabilities [96]. Human PDAC-resected tissues overexpressed GRP78, IRE1α, and XBP1 [97]. Spliced XBP-1 (XBP1s) responds to LC3B transcription [98], a genuine marker of autophagy activation. In breast cancer cells (MCF-7 and MDA-MB-231), ER stress induces tunicamycin (TM) insensitivity through the upregulation of GRP78, IRE1α, and LC3. Simultaneously, it inhibits autophagy via 3-methyladenine (3-MA), and enhances the sensitivity of TM-induced apoptosis [99]. Furthermore, in an investigation of an IRE1α-PERK-ATF6-deficient ER stress model, autophagy was reduced in the IRE1α signaling pathway but not in the PERK and ATF6 signaling pathways [100]. Additionally, the IRE1α kinase domain is required for autophagy regulation, which activates the IRE1α-JNK signaling pathway [101]. Apoptosis signal-regulated kinase (ASK1) is a downstream signaling molecule of TRAF2 that activates c-Jun *N*-terminal protein kinase, which is triggered to maintain ER homeostasis by autophagy [97,102]. Cancer cells survive by inducing p62-dependnt canonical autophagic degradation to overcome ER damage [103]. For example, ER stress inducers such as TM, brefeldin A, and thapsigargin (TG) activate autophagy in colon and prostate cancer cells, effectively protecting cells against ER stress-induced cell death [104].

LC3II and ATG12-ATG5 conjugates largely support the fundamental processes of autophagy and activating the transcription of the relevant autophagy genes is critical in maintaining the autophagic flow. The data show that the UPR PERK branch regulates these genes during ER stress [105]. PERK is essential in ER-mediated autophagy. Previous reports indicate that the accumulated polyglutamine (poly Q) proteins in cytosol reduce the function of the proteasome, leading to the activation of autophagy by inducing the PERK UPR branches [106,107,108]. In support, several investigations on cancer suggest the development of autophagy via the stimulation of the PERK UPR branches. For example, ER stress promotes mitochondrial dysfunction by activating the eIF2 [109]. In addition, PERK is more responsible for ER adaptive response by inhibiting DNA damage, inflammation, and genomic stability during radiotherapy [110,111,112]. These data indicate that cancer cells are resistant to moderate and sustained ER stress when PERK is activated. This adaptation is regulated by the phosphorylation of eIF2*α* and stimulates autophagy via ATF4-dependent ATG12 and pseudokinase TRB3 expression, which inhibits the Akt/mTORC1 axis [113]. Moreover, PLX4720 is a BRAF inhibitor that activates ER stress-induced autophagy flow in melanoma cells by activating the PERK pathway [114]. ATG5 knock-out cancer cells show a higher response to ER stress [115], which indicates that the suppression of autophagy amplifies the degree of ER stress. However, autophagy is also responsible for cancer cell apoptosis as BRAF inhibitor increases apoptosis by activating ER stress-mediated autophagy [114]. Multiple investigations reveal that the induction of autophagy in cancer may be beneficial in treating cancer. In one of the investigations, HA15, an antimelanoma drug, was used to stimulate apoptosis and autophagy by targeting GRP78 [116]. Autophagy is accompanied by vesicle aggregation, the transition of LC3I to LC3II, and the production of autophagosomes. The therapeutic efficacy of HA15 in melanoma cells decreased with lowered autophagy and apoptosis, suggesting that autophagy may suppress malignant neoplasms [116]. The PERK branch of UPR induces autophagy, which plays several regulatory roles in the cannabinoid-dependent survival of cancer cells [113]. Thus, targeting the PERK signal may be the key to autophagy and ER homeostasis-mediated cell survival, where the PERK-eIF2α pathway constitutes a potential approach to overcome the hurdles while dealing with malignant tumors.

The ATF6 branch in the UPR is poorly understood in terms of ER stress and autophagy. However, it is known that ATF6 transcriptional activity is linked to the activation of autophagy via HSPA5 overexpression and the consequent downregulation of AKT1/AKT [117]. Previous reports indicated that the ATF6 fragments enhance nuclear translocation in multiple cancers, including HCC and Hodgkin’s lymphoma [118,119]. Moreover, in a clinical trial, 50% of patients with high nuclear ATF6 levels had a poor histological response to treatment [120], suggesting the role of ATF6 expression during metastasis. In ovarian cancer, the signal transducer and activator of transcription 3 (STAT3), an ER stress inducer, is activated through ATF6. Thus, ATF6 mediates autophagy via LC3B upregulation for cancer cell survival, but ATF6 down-regulation could reverse dormant cell resistance in vivo. These observations indicated that ATF6 has a synergistic effect on autophagy for cancer cell survival. Furthermore, the ATF6-related transcription factor CEBPB (CCAAT/enhancer-binding protein beta) has been connected to *Interferon Gamma* (IFNG)-dependent autophagy via DAPK1 expression (death-associated protein kinase 1) [121]. The suppression of ATF6 or the Ras homolog enriched in brain (Rheb) reinstates rapamycin resistance in dormant tumor cells, revealing the regulation of autophagy in tumor cells via the ATF6-Rheb-mTOR pathway [122]. ATF6 activation dramatically enhances the rapamycin resistance in malignant osteosarcoma (OS) [120]. This can be achieved by reducing Bax activation, which inhibits Rheb-mTOR signaling pathway [123]. In osteosarcoma and NIH3T3 cells, the ULK/ATG13/FIP200 complex is required to induce autophagy as these proteins regulate autophagy through enhanced mTORC1 phosphorylation [124,125].

Various conditions such as starvation, hypoxia, oxidative stress, protein aggregation, and increased ER stress activate AMPK. AMPK triggers the autophagic process through the dephosphorylation of mTORC1 and the activation of ULK. ULK acts in conjunction with FIP200 and ATG13, which phosphorylates Beclin 1, which leads to the activation of VPS34 and phagophore formation [22]. Consequently, it activates the nucleation of the phagophore by phosphorylating components of the PI3KC3 complex I (consisting of class III PI3K bind with ULK1 complex). PI3P recruits the WIPI2 (WD repeat domain phosphoinositide-interacting protein 2) and DFCP1(zinc-finger FYVE domain-containing protein 1) that interact with their PI3P-binding domains. WIPI2 directly binds with ATG16L1 and recruits the ATG12~ATG5–ATG16L1 complex [126] enhancing the ATG3-mediated conjugation of ATG8 family proteins (ATG8s), which are essential for the elongation and closure of the phagophore membrane and finally merging with the lysosome. Eventually, acidic hydrolases in the lysosome break down the autophagic cargo, releasing nutrients back into the cytoplasm for reuse by the cell. Moreover, in selective autophagy, LC3, GABARAPs, and FIP200 critically involve LIR and FIP motifs binding with ER-phagy receptors sequestration of specifically labeled ER into autophagosomes release the ER stress. Figure 1 illustrates the interaction of ER stress and autophagy.

## 7. Noncanonical Stress Response Induces Autophagy and Promotes Cancer Progression

Noncanonical ER stress acts as an additional route for the activation of the UPR and covers multiple avenues, potentially influencing cell fate in ways distinct from the basic UPR. Recently, multiple novel pathways linked cell stress to the UPR and autophagy as noncanonical ER stress responses [127]. Noncanonical stress response leads to the transcription of several autophagy genes and the upregulation of autophagic flux, which emerged as a stress adaptation mechanism in cells [128]. Furthermore, noncanonical ER stress activates cytoprotective autophagy associated with angiogenesis, endothelial cell proliferation, and migration [127]. Here we discuss the ISR, DNA damage response, ER calcium, ER-phagy, and other pathways that are just garnering needed attention with recent scientific advances. Our report focuses on the cytoprotective effects of autophagy and noncanonical ER stress response mechanisms. Cancer therapy can be improved by targeting the UPR or noncanonical ER stress mechanism by blocking cytoprotective autophagy. Table 1 lists apoptotic marker proteins and stress inducers in canonical and noncanonical ER stress responses in multiple cell lines.

## 8. ER-Specific Autophagy

Cancer cells trigger the UPR as an adaptive mechanism in the presence of stress components in the tumor microenvironment (TME). Recently, ER-specific autophagy has been associated with the removal of damaged ER and restoring ER homeostasis [51]. Peter Walter’s group first described ER-phagy and demonstrated the selective engulfment of ER into autophagosomes, a necessary process for the survival of cells exposed to severe ER stress [138]. The precise process of ER-specific autophagy in cancer is complex, but its role in many malignancies has been uncovered. However, some studies revealed that the role of reticulophagy receptors mediated ER stress tolerance in tumorigenesis [48]. ER-phagy receptors such as SEC62, CALCOCO1, CCPG1, TEX264, FAM134B, ATL3, and TRIM13 are localized in different ER parts and maintain ER homeostasis via different molecular mechanisms. Several reports indicated cancer cell death via ER stress inducers. For example, brigatinib is one of the significant ER stress inducers. Zhang et al. reported that brigatinib treatment induced autophagy in a dose-dependent manner, as indicated by high LC3B-II conversion and levels of Beclin 1, Atg5, and Atg7 [139]. Stimulation with brigatinib enhanced the interaction between Beclin 1 and Atg14L and reduced its association with Bcl-2. Additionally, endogenous LC3B and GFP-LC3 puncta formation were greatly enhanced in colorectal cancer (CRC) cells treated with brigatinib [139]. However, ER stress induced by brigatinib triggers an autophagic response via Fam134B, which acts as a protective mechanism to alleviate excessive ER stress in human CRC [140,141]. Additionally, FAM134B is shown to have a cellular protective mechanism through the degradation of ruptured ER-side, revealing oncogenic characteristics in ESCC (esophageal squamous cell carcinoma) [141].

Nevertheless, the loss of ER-phagy receptor FAM123B increases ER stress and reduces cell proliferation in a breast cancer xenograft model [103]. Limited information on FAM134B necessitates further investigation to identify the precise protective mechanism. Moreover, SEC62 is a crucial molecular component in sustaining and retrieving ER homeostasis, and increased ER calcium regulates stress resistance [142,143]. SEC62 overexpression is associated with reduced survival and progression-free survival in cancer patients. Additionally, higher SEC62 levels are linked to larger tumor size, tumor ulceration, and distant metastases in the various cancer types [144,145]. Still, it is unclear how these receptors are activated or what is the nature of their involvement in parts of the ER homeostasis, i.e., what drives them to bind to LC3 at a particular time and how do they distinguish intraluminal cargo. However, severe stress or starvation response to the degradation of ER via ER-phagy receptors has a critical role in the cell death mechanism. This excessive ER degradation by FAM134B and TEX264 causes cell death in glioblastoma [146,147]. CCPG1 is downregulated in retinoblastoma cells under stressful conditions and linked to a poor prognosis in cancers [148]. Furthermore, RTN3, an ER-phagy receptor, is highly downregulated in hypoxic conditions, but its overexpression triggers tumor necrosis factor-α (TNF-α)-related apoptosis-inducing ligand (TRAIL) and the Fas-dependent apoptosis of renal cancer cells compared with normal cells [149]. The role of ER-phagy is still under debate despite accumulating evidence suggesting the critical role of ER-phagy in cancer. However, autophagy provides stress tolerance and preserves cell viability under hostile environmental conditions. Further insights are necessary to determine whether targeting ER-phagy should be considered or not. If considered, this might open a new door for anticancer therapy. Table 2 lists anticancer drugs that modulate ER stress-mediated autophagy.

## 9. ER Ca^2+^ Regulates Autophagy

The calcium signal is a critical regulator of diverse cellular events over multiple timelines, including hallmark events such as proliferation, gene transcription, and invasiveness [164,165]. Many investigations have shown that calcium signals enhance cancer progression in various cancer types by activating autophagy in cancer cells, but whether Ca^2+^ regulates autophagy in cancer is still up for debate. Still, we made an extensive effort to address how autophagy regulates calcium signaling and influences cancer progression. Ca^2+^ regulates autophagy in different pathways, including the calcium–ERK and CaMKKβ–AMPK–mTOR pathways [166,167]. Further, increased cytosolic Ca^2+^ ([Ca^2+^]_i_) in response to ER stress stimuli activates LKB–AMPK, and DAPK-1, which execute a cell defense mechanism by regulating calcium/calmodulin-dependent protein kinase kinase (CaMKK) β-mediated autophagy [168,169,170]. Thus, autophagy regulated by [Ca^2+^]_i_ induces Beclin 1, ATG5, and LC3B expression in gastric cancer cells [163]. Jeremi Laski et al. observed that AMPK signaling facilitates autophagy activation in a spheroid model of epithelial ovarian cancer (EOC) metastasis. CAMKK inhibitor effectively limits AMPK activation and autophagic flux in EOC spheroids, resulting in decreased cell survival [171]. Calcium signals induce ROS in mitochondria, which cause STAT3 to be phosphorylated and translocated into the nucleus [172]. STAT3 is a transcription factor that stimulates multiple autophagy-linked genes. According to the Maycotte study, STAT3-mediated autophagy rapidly proliferates cancer cells [173]. Thus, autophagy inhibition could be a potential treatment for triple-negative breast cancers. Moreover, a recent study found that the mitogen-activated protein kinase (MAPK/ERK) signaling pathway is associated with calcium signals. It protects PDA cells from the harmful effects of the KRAS pathway and induces a cellular recycling process by autophagy [174].

Calcium signaling is essential to master regulators in cancer, and JNKs are protein kinases that are critical for stress signaling pathways. Under ER stress, the activation of autophagy contributes to resistance in several anticancer therapies. For instance, ER stress-induced calcium release leads to the activation of calmodulin kinase II (CaMKII), which induces JNK phosphorylation and LC3-II expression. Evodiamine is an intracellular calcium scavenger that promotes autophagy in glioma cells via the calcium–JNK signaling pathway [175]. Furthermore, [Ca^2+^]_i_ is utilized as a signaling molecule to regulate several cellular events. Zhao et al. revealed that Ca^2+^ activates the EndoU family proteins to modulate the formation of tubular ER networks and contribute to dynamic ER shaping [4]. This Ca^2+^-dependent ER remodeling is susceptible to cancer. The human protein atlas analysis indicates that aberrant expression of EndoU reduces the survival rate in cancer patients. Additionally, the transmembrane BAX inhibitor motif-containing 6 (TMBIM6) is a Ca^2+^ channel-like protein that acts as an autophagy inducer and a known novel modulator of ER-stress-induced apoptosis [176]. The cellular protective effects of TMBIM6 are first described by Chae et al., where protective effects were attributed to enhanced PRKAA phosphorylation, the regulation of MTORC1 activity, and the activation of PPP3/calcineurin, stimulation of TFEB (transcription factor EB) nuclear translocation that enhances autophagy flux in cancer cells [177,178,179]. When TMBIM6 binds to the inositol 1,4,5-triphosphate receptor (IP3R), it reduces mitochondrial Ca^2+^ uptake and disrupts cellular ATP, triggering AMPK-dependent autophagy [180]. However, an increase in [Ca^2+^]_i_ could be cytotoxic under specific conditions through the activation of caspase-induced apoptosis. Neuroblastoma cell death is inhibited by preventing Ca^2+^ release from the ER by blocking IP3 channels or chelating cytosolic Ca^2+^ using a cell-permeable Ca^2+^ buffer [181]. Collectively, the high possibility of ER stress induces autophagy and modulates Ca^2+^ release from the ER, optimizing the cancer cell sensitivity in a context-dependent manner.

Additionally, DNA damage response (DDR) influences the cell cycle. The cellular DDR system continuously inspects the genome and is ready to disseminate several signals depending on the detected damage. Autophagy is one such system identified by DDR and a potential repair system. DDR inhibits SKN-1 and activates IRE1α, FOXO3a, ATM, and P53 to induce autophagy and enhance cell survival [1]. Furthermore, the integrated stress response (ISR) promotes cell survival signals by negatively regulating apoptotic pathways through the phosphorylation of eIf2α by GCN2, HRI, and cMYC. ATF4 and eIF2α are common features of autophagy activation.

## 10. DNA Damage Response (DDR)

DNA damage is triggered by multiple factors, including endogenous metabolites, antiinflammatory drugs, environmental and dietary carcinogens, and genotoxic cancer treatments. DNA damage activates complex signals to control cell fate, promoting DNA repair with cell survival or apoptosis. DNA repair, DNA damage tolerance, and factors influencing the activation of UPR and autophagy are indicators of cell survival [75,182]. DDR has a crucial function in cancer initiation and the progression of cancer. Furthermore, DDR prevents mTORC1 signaling, ATG1/ULK1 activation, and the Beclin 1 complexes, accomplished via DNA damage sensors, including FOXO3a, ATM, ATR, and p53 [183]. The connections between autophagy and the DDR-mediated anticancer barrier are unknown.

Furthermore, the DDR-autophagy axis is triggered by DNA damage, which is essential for several functional outcomes of DDR signaling. The DDR-autophagy axis also plays a vital role in preventing radiation or chemotherapy-induced cell death by repairing DNA damage [184]. Moreover, an ataxia–telangiectasia mutation (ATM) functions upstream of p53 and modulates the DDR pathway necessary for fixing double-stranded DNA breaks. Previous investigations revealed that the ATM is activated by several noncanonical modes of cellular stress, such as DDR, redox signaling, and cancer [185]. ATM inhibits the autophagy suppressor mTORC1, leading to autophagy activation and the repair of damaged DNA [184]. Thus, ATM plays a protective mechanism by suppressing different apoptotic pathways and implicates an autophagy-mediated proteostatic response by AMPK-like signaling pathways [186]. Moreover, a meta-analysis revealed the association of ATM gene variants with breast cancer [187]. Forkhead box protein O3 (FOXO3) potentially activates ATM and influences DNA repair [188,189]. Additionally, FOXO family proteins are homeostasis regulators that control apoptosis regulated by UPR signals [190]. These UPR signals control the expression of genes involved in autophagy, cellular metabolic activity, and cancer growth and metastasis. For instance, nuclear FOXO3 contributes to doxorubicin chemotherapy resistance with a poor prognosis in neuroblastoma [191]. Additionally, p53 activation causes the production of the phosphatidylinositol phosphate PTEN and the lysosomal protein DNA damage-regulated autophagy modulator protein 1 (DRAM1) [192]. This DRAM1 expression by p53 has been reported to enhance the migration and invasion abilities via autophagy in hepatoblastoma cells [193]. Additionally, DRAM1 is expected to have a controversial discourse in dealing with the increased autophagy by DNA damage. Moreover, DDR suppresses SKN-1 activity, which increases unsaturated phosphatidylcholine (PC), activating the IRE-1α/XBP-1 branch of the UPR-mediated adaptive cellular response [1]. Thus, unsaturated fatty acids are required simultaneously for DNA damage and UV-induced ER stress resistance, which promotes cancer progression. Separately, the maintenance of ER function is primarily unknown.

Current evidence indicates that DNA damage potentially induces autophagy to safeguard mechanisms and turnover cellular homeostasis [194]. Activation of autophagy via upregulation of proteins such as Beclin 1 and Atg7 delay apoptotic response to DNA damage in noninvasive breast cancers [195]. However, unresolved damage is harmful and triggers cell death events such as apoptosis and necrosis, which also participate in pathways linked to tumor suppression. For example, DNA-targeted chemotherapeutic agent carboplatin becomes resistant to DNA damage repair pathways. However, it increases the cytotoxic effect with autophagy inhibitor chloroquine in breast cancer stem cells [196]. All of the evidence indicates autophagy as a survival strategy at modest levels of DNA damage. Figure 2 illustrates the molecular pathways that contribute to cell survival during stress conditions, indicating potential therapeutic targets for cancer.

## 11. Integrated Stress Response (ISR)

In response to numerous stress events, eukaryotic cells activate an adaptive pathway known as ISR that restores cellular equilibrium. This potentially activates HRI, PKR, PERK, and GCN2 kinase, and UPR pathways converge on the phosphorylation of eIF2α at Ser51, the core of ISR [127,197,198,199,200]. The phosphorylation of eIF2α reduces global protein synthesis while facilitating the translation of specific genes, such as ATF4, which cellular recovery through autophagy [199,201]. However, the role of ISR transducers in tumor initiation or cellular homeostasis in most cells is still unknown. Furthermore, oncogene activation can promote ISR-mediated pro-survival autophagy in cancer biology. The ISR and protein degradation pathways play critical roles in cell proliferation, metastasis, and chemotherapeutic resistance. In response, the PERK-mediated phosphorylation of eIF2α induces LC3 puncta by activating ATF4, promoting the disposal of misfolded and protein aggregates via autophagy during tumor growth [202]. The expression of autophagic genes such as LC3-II and BECN1 correlates with the cytoprotective role in promoting cell survival in fibrosarcoma, lung, cervical carcinoma, and HCC [127,147,203,204,205,206,207,208]. The PERK is activated by proto-oncogene c-Myc and N-Myc under amino-acid starvation [127,209,210]. The aggregation of misfolded proteins causes proteotoxicity, accelerating cell death by inhibiting ISR-mediated eIF2α and ATF4, a common feature of lung adenocarcinomas [211]. According to these findings, c-Myc overexpression-associated malignancies may be susceptible to UPR inhibition.

Furthermore, GCN2 activation is reported in human keratinocytes, tumor cells, and mouse embryonic fibroblast (MEF) cells in response to chronic glucose deprivation. However, there may be a collateral implication arising from using AA as an alternate energy source in the absence of glucose. Therefore, tumor progression needs proteostasis adjustment by the upregulation of ATF4 and the modulation of cellular stress responses. Recent investigation suggests that GCN2-mediated autophagy prevents proteotoxicity in breast cancer cells and demonstrates complementary mechanisms to ensure survival [150]. ATF4 or GCN2 expression abrogation significantly inhibited tumor growth in MEFs [60]. RNA-dependent protein kinase is essential in the eukaryotic response to viral infection. Simultaneously, PKR is involved in cellular differentiation, growth control, and apoptosis. Multiple observations suggested that PKR protein expression is high in several cancer cells and low in normal cells; however, its role in human cancer is poorly understood [212,213]. PKR controls misfolded protein clearance in breast cancer by suppressing its release through exosomes and accelerating lysosomal breakdown [151]. However, PKR-depleted cells and PKR/MEFs are vulnerable to autophagy inducers such as STAT3 inhibitors and poly (I:C). This specific finding indicates that PKR is necessary for autophagy activation. Chang et al. determined that PKR activation is essential for imiquimod (IMQ)-induced autophagy [151]. The above mechanisms establish that ISR-mediated autophagy triggers chemotherapy resistance in breast cancer. Thus, pharmacological alterations in ISR could form a potential novel therapeutic strategy.

## 12. The Consequence of Unresolved ER Stress to Cell Death Mechanism

Many stimuli that modify cellular homeostatic function can activate ER stress and its mediated downstream apoptosis. Pathways such as UPR-mediated mitochondrial apoptosis, intracellular autophagosomal platform, and ER stress-induced death receptors influence ER stress-mediated cell death [214]. The UPR sensor is broadly related to pro-survival pathways in response to ER stress by regulating multiple chaperones. During chronic or unresolved ER stress, UPR activity is enhanced and leads to a pro-apoptotic pathway. The PERK phosphorylate eIF2α specifically induces the transcription factors ATF4 and CHOP, which play a key role in cell death. For example, the adenoviral administration of ATF4 was strong enough to cause trigger apoptosis in MEFs, and CHOP overexpression enhanced apoptosis [215]. This eIF2α/ATF4/CHOP response to ISR and other stimuli, such as redox imbalance and nutrient deprivation, can activate other kinases.

CHOP and ATF4 work synergistically to stimulate apoptosis, although how this is accomplished is still unclear. ROS plays a vital role during cell death irrespective of apoptotic or necrotic conditions mediated by the ATF4/CHOP axis. ER oxidoreductase 1 (ERO1a) transcription is induced by CHOP, which hyperoxidizes the ER environment. In support, a rescue experiment with *Caenorhabditis elegans* cells revealed the prevention of TM-induced cell death by ERO1a knockdown, indicating the significance of ERO1a during apoptosis [216]. ATF4, ATF6, and XBP1 use CHOP as a transcriptional target. However, CHOP-driven cell death does not indicate the participation of the PERK-ATF4 pathway. Contrastingly, the overexpression of CHOP itself is not always sufficient to kill cells [217]. JNK regulates apoptotic pathways and is involved in necrosis in response to ER stress components. Hence, the JNK branch of the IRE1α pathway promotes cell death. Furthermore, mitochondrial apoptosis involves pore formation on the mitochondrial outer membrane, and the activation of pro-apoptotic BH3 regulates Bcl-2 family members Bax and Bak [218,219]. Contrastingly, antiapoptotic Bcl-2 family proteins inhibit pore formation.

ER stress-mediated autophagy is always a matter of debate in ER stress investigations. Autophagy involves autophagosome synthesis and lysosomal breakdown, which decrease proteotoxicity. However, autophagosome accumulation is subsequently unfused to lysosomes, directly inducing cellular toxicity [220]. In chronic ER stress cells without caspase-9 or Bax and Bak, cell death depends on caspase-8 as it is the apical protease in the cascade [221]. According to different research groups, prolonged ER stress or associated proteotoxic stress activated caspase-8 on an intracellular autophagosomal platform. Sequestosome 1/p62 and LC3 are two major autophagy regulators that cause caspase-8, FADD, and Atg5 to be recruited to autophagosomal membranes. Treatment with TM promotes the synthesis of the intracellular death-inducing signaling complex (iDISC) or stressosome, which results in caspase-8-dependent apoptosis in colon tumors and breast cancer cells [222]. ATG5 and ATG7 were knocked out to prevent cell death induced by caspase-8. Further, it was identified that Atg5, FADD, and pro-caspase-8 complex coordinate caspase activation and cell death [223,224]. Autophagosomal membrane stimulated by ER stress inducers forms a kind of recruitment force and activates caspase-8 to trigger apoptosis. However, further investigations are necessary to determine the ability of cells against ER stress. Therefore, autophagosome accumulation may represent a therapeutic strategy for tackling disease progression.

## 13. Maximize the Therapeutic Benefit by Manipulating ER Stress and Autophagy

Autophagy acts as an additional hurdle when a patient is under the influence of chemotherapy; it establishes cytoprotective mechanisms along with lethal programs dependent on chronic ER stress, creating a microenvironment for rapid tumor development through energy generation and other processes [225]. The cellular protective effects lead to tumor progression, and autophagy is responsible for decreasing the patient’s survival in a stress-dependent manner and developing drug resistance (Table 1). Intriguingly, ER stress is genuinely associated with apoptosis, and understanding cells’ behavior and reactions to ER stress facilitates the development of novel cancer therapies. Mild-chronic stress is significantly correlated with autophagy induction with survival. However, chronic stress reflects ER-stress-mediated apoptotic cell death with autophagy inhibition. Consistent developments in research suggest that ER stress can be mitigated by blocking autophagy using genomic interference against autophagic genes or pharmacological inhibitors of autophagic components. For instance, 3-methyladenine (3-MA), LY294002, ABT-737, resveratrol, chloroquine (CQ), bafilomycin A1 (Baf A1), and hydroxychloroquine (HCQ) are standard autophagy inhibitors that inhibit PI3K in early autophagy and block vacuolar-type H (+)-ATPase in late autophagy [226]. Still, different output suggests that the autophagy inhibitors such as CQ, 3-MA, HCQ, Baf A1, elaiophylin, 4-acetylantroquinonol B, thymoquinone, and S130 intersect with ER stress (Table 3). Vu et al. determined that the use of autophagy inhibitors or knockdown of the autophagy-related genes such as BECN1 and ATG7 enhanced ER stress-mediated tumor cell death [227]. Despite these insights, critical concerns remain on the specific stage of autophagy that needs to be sensitized to develop anticancer therapy.

Growing investigations indicate that targeting autophagy with anticancer drugs enhances apoptosis and could be an effective therapy against various cancer types. For instance, brigatinib activates ER-phagy, a pro-survival mechanism. The inhibition of late-stage autophagy by CQ increases autophagosome accumulation by rupturing the lysosomal membrane and increases ER stress-mediated apoptosis, improving the efficacy of CRC cell response to brigatinib in vitro and in vivo [139]. However, some ER stress responses, such as Ca^2+^ and ER-phagy receptors, play a defensive role. For example, verotoxin-1 (VT-1) response to UPR sensors such as IRE1 and ATF6 increased apoptotic cell death with autophagy. However, the protective mechanism of ER-phagy against excessive ER stress depends on the type of cell line used. Due to the dual roles of VT-1, it is necessary to understand whether it is suited for combination therapy, where it is supposed to protect normal cells and kill tumor cells. Thus, future investigations are necessary for understanding the precise mechanism. Pharmacological ER-phagy receptor regulator vitexin binds with BIP inhibiting ER-phagy and suppressing tumor progression. On the other hand, it acts synergistically with ER stress inducers to slow down cell proliferation [103]. Further, Hart et al. demonstrated the vital role of PERK in autophagy induction, where it enhances the expression levels of c-Myc-induced autophagy by increasing LC3 I/II conversion for autophagosome formation [209]. Meanwhile, the suppression of autophagy with Baf A1 in P493-6B cells enhanced cell death. Thus, combining an autophagy inducer with an autophagy inhibitor will be more effective for anticancer therapy. Table 4 lists the efficacy of various ER stress inducers and autophagy inhibitors in cancer therapy. However, emerging evidence indicated that hyperactive autophagy could function as a fatal mechanism resulting in autophagy-dependent cell death under specific physiological and pathological conditions.

The acceleration of cell death via the pharmacological modulation of cellular responses toward excessive ER damage by ER-phagy may prevent tumor development and growth. For instance, ER stress induces FAM134B, LC3, and Atg9 expressions, which together mediate excessive ER-phagy via the tribbles homolog 3–dependent (TRB3-dependent) inhibition of the Akt/mTORC1 axis in glioma and Hela cells (Table 4). In addition, Ca^2+^ transfer from the ER to the other intracellular compartments is crucial for the regulation of cell survival and cell death. For instance, Ca^2+^ chelation or CaMKK inhibitor treatment with nigericin synergistically suppresses cellular spheroid formation. Vu et al. proposed that calcium-mobilizing compounds combined with autophagy inhibitors are the more prominent therapeutic strategy to treat patients with glioblastoma [227]. Furthermore, high [Ca^2+^]_i_ is also responsible for autophagy and ROS-mediated cell death (Table 4). This controversial role of Ca^2+^ signaling could be solved in a context-dependent manner, necessitating further investigation into the autophagic role in Ca^2+^ signaling. Furthermore, data suggest that tumor cells use autophagy to withstand radiation or chemotherapy-induced cell death. Additionally, the genetic or pharmacological suppression of autophagy with ER stress inducers sensitizes malignant cells to genotoxic treatment in in vitro and in vivo models. Figure 3 describes the influence of chemotherapy on cell fate during adaption and severe ER stress conditions.

## 14. A Comprehensive Patient’s Data Analysis for Accelerating Cancer Research and Precision Medicine

We used bioinformatics analysis to investigate the transcriptional expression of proteins as they are potential indicators of patient survival in various cancers and drug development. Additionally, bioinformatics analysis enables detailed proteomic and genomic analysis. Moreover, bioinformatics tools were used to mine expression data and perform subsequent data analyses to support the suggested notion. These specific analyses were based on the cancer genome atlas (TGCA) to extrapolate the molecular mechanisms, interactions, and linked biomarkers. Comprehensive data analysis revealed the thoughtful bridge between ER stress and autophagy markers. Protein–protein interactions (PPIs) are essential in many biological processes. Their dysfunction has been correlated to the pathogenesis of several diseases, including cancer. These transient or permanent, identical or heterogeneous, and specific or nonspecific interactions could assist in developing novel drugs against cancer, and we used the STRING database to present the PPI based on a list of UPR sensors and their downstream ER-phagy receptors, calcium-regulating proteins, kinases, and enzymes. It was observed that the UPR sensor regulates downstream proteins, and TMBIM6 transmembrane protein has higher interaction with other proteins than CAMKK2 kinase, plasma membrane-bound FAM134B, and RTN3, CCPG1 SEC62 (ER-phagy receptor). Understanding these PPIs provides insights into cellular activities and facilitates disease diagnosis and drug development.

Furthermore, Database for Annotation, Visualization, and Integrated Discovery (DAVID) bioinformatics tools show that high PPI triggers a cellular stress response and accelerates autophagy. Cancer cells have adapted to utilize autophagy as a survival mechanism [239]. Yihua Wu and colleagues demonstrated the involvement of 137 genes in autophagy-linked pathways using ATdb expression data from 25 different cancer types [240]. Further, an investigation by Lindstrom et al. suggested that patients with high autophagic proteins have a poor prognosis for survival in the lung, prostate, colorectal, stomach, liver, breast, colorectal, lung, and cervix [241]. Here, we used three sets of patients’ data from TGCA. Our analysis demonstrated a positive correlation between fam134B, CCPG1, TEX264, and Camkk2 genes with patient survival. However, the high expression of TMBIM6, ATF4, EndoU, RTN3, CKAP4, and XBP1 reduces the cancer patient survival rate. The Venn diagram shows the correlation between molecular processes and the selected group. In this analysis, Group 1 (S1) is linked to UPR sensor and downstream processes, Group 2 (S2) is linked with ER-phagy receptor, and Group 3 (S3) is linked to calcium-regulated proteins that have high-impact mutations. There were 1070 cases of overlap between the three groups. However, Endou, CCPG1, TEX264, and ATL3 did not have any meaningful relationships (Figure 4 and Figure 5). Further investigations are warranted in this respect.

## 15. Concluding Remarks, Open Question, and Future Perspective

Cancer cells evolve multiple mechanisms to withstand numerous stresses. Tumor cells utilize degraded components to generate energy through autophagy and grow efficiently under stressful conditions. However, the autophagy genes that are involved in these pathways are challenging to regulate, making the cancer difficult to drug. Thus, aiming at UPR or noncanonical ER stress programs is a viable option for the development of novel cancer therapy. Additionally, it is suggested that selective autophagy suppression in cancer cells and systemic autophagy stimulation may be coupled to improve the efficacy of anticancer therapy. Our analysis shows that the high expression of canonical and noncanonical components negatively regulates survival rate. However, the explanation of the regulation mechanism by autophagy is at the immature stage. Unresolved ER stress negatively affects cancer cells’ survival rate through autophagy inhibition, but it is unclear which specific stage of autophagy inhibition is more effective. Altogether, UPR and its components are crucial in reducing tumor heterogeneity. However, specific context-dependent components responsible for ER stress recovery could be more precise. Further detailed mechanistic investigations are necessary. ER stress can be presented as a plausible model with autophagy as a target. Still, challenges and opportunities remain open in cancer therapy.

## Figures and Tables

**Figure 1 cells-11-03773-f001:**
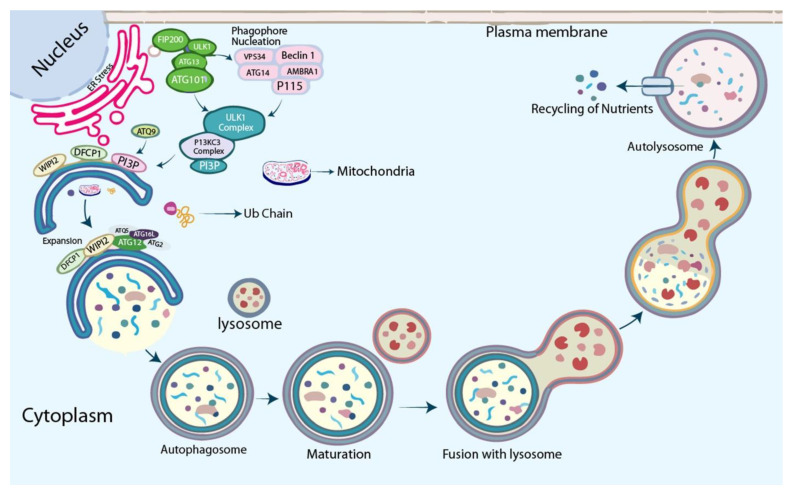
Graphical model illustrating the interaction between ER stress and autophagy. Multiple conditions, such as increased ER stress, oxidative stress, starvation, hypoxia, and protein aggregation, activate AMPK. AMPK initiates the autophagy process by dephosphorylating mTORC1 and activating ULK. Further, the ULK/ATG13/FIP200 complex is necessary for inducing autophagy because these proteins regulate autophagy by increasing mTORC1 phosphorylation. Additionally, ULK acts in conjunction with FIP200 and ATG13 to produce a complex that phosphorylates Beclin 1, resulting in VPS34 activation and phagophore assembly, then activates nucleation of the phagophore by phosphorylating class III PI3K (PI3KC3). PI3P draws the PI3P effector proteins WIPI2 and DFCP1 (zinc-finger FYVE domain-containing protein 1) that interact with their PI3P-binding domains. WIPI2 directly binds with ATG16L1 and recruits the ATG12~ATG5–ATG16L1 complex enhancing the ATG3-mediated conjugation of ATG8 family proteins (ATG8s), essential for elongation and closing of the phagophore membrane. Next, acidic hydrolases in the lysosome break down the autophagic cargo, and reclaimed nutrients are transferred to the cytoplasm for reuse by the cell.

**Figure 2 cells-11-03773-f002:**
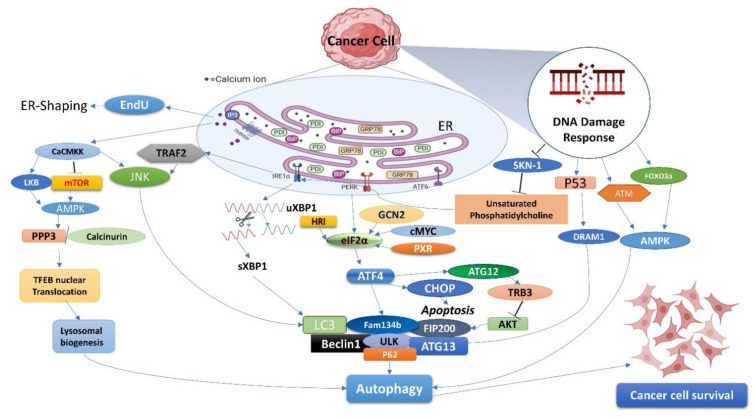
Graphical representations of molecular pathways that contribute to cell survival during stress conditions suggest potential cancer therapeutic targets. ER stress in cancer cells is triggered by multiple factors, effectively inducing autophagy. Canonical pathway is linked to a common UPR pathway involving IRE1α, PERK, and ATF6. Here, the pathway regulates the LC3, Beclin 1, and P62 ATG proteins and shows enhanced cell viability. Contrastingly, the noncanonical pathway involves Ca^2+^, ER-phagy, DNA damage response, and ISR-mediated autophagy. Under stress conditions, Ca^2+^ release from ER by Ca^2+^ channel and TMBIM6 activates EndoU for ER-shaping and CaCMKK-mediated autophagy by activation of JNK and AMPK. AMPK regulates PPP3/calcineurin, stimulating TFEB (transcription factor EB) nuclear translocation and influencing lysosomal biogenesis. Separately, the DDR-autophagy axis is one of the most intriguing functional consequences of preventing radiation or chemotherapy-induced cell death by repairing DNA damage. DNA damage response inhibited SKN-1 and activated IRE1α, FOXO3a, ATM, and P53-mediated autophagy, enhancing cell survival rate. In addition, ISR promotes cellular survival signaling by negatively regulating cell death pathways through the phosphorylation of eIF2α by GCN2, HRI, and cMYC. The eIF2α and ATF4 are common features of autophagy activation.

**Figure 3 cells-11-03773-f003:**
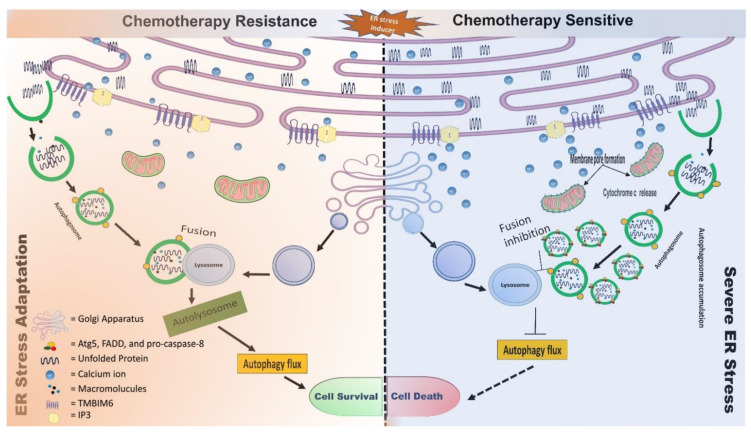
Schematic illustration of the effect of chemotherapy on cell fate during adaptation and severe ER stress conditions. Chemotherapy activates ER stress-mediated autophagy, which activates a protective mechanism for cancer cells during chemotherapy. On the other hand, autophagy inhibition executes apoptosis through multiple pathways, such as mitochondrial apoptosis, UPR, and autophagosomal apoptosis. Autophagosome accumulation in cytosol increases ER stress. Additionally, chronic ER stress increases mitochondrial Ca^2+^ imbalance, which ruptures the mitochondrial membrane; activates Bax, Bak, and caspase 9; and releases cytochrome c-mediated cell death.

**Figure 4 cells-11-03773-f004:**
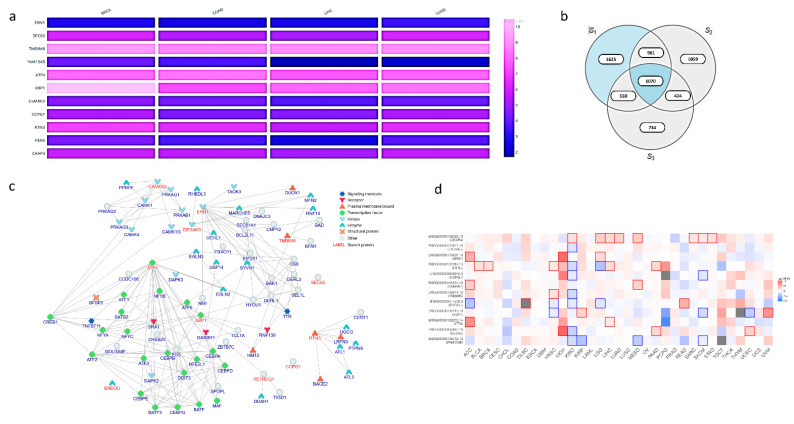
Current understanding of molecular components linked to ER stress and autophagy that influence tumor cell survival and their clinical relevance. (**a**) Expression profiles of ER stress proteins, ERphagy proteins, and Ca^2+^ signaling proteins in TCGA cancer data (by GEPIA2). (**b**) Venn diagram of ER stress proteins, ER-phagy proteins, and Ca^2+^ signaling proteins in TCGA cancer data (by NIH GDC data portal). (**c**) Protein–protein interaction data of ER stress proteins, ER-phagy proteins, and Ca^2+^ signaling proteins (inBio Discover). (**d**) Survival map of ER stress proteins, ER-phagy proteins, and Ca^2+^ signaling proteins associated with different TCGA cancer (GEPIA2).

**Figure 5 cells-11-03773-f005:**
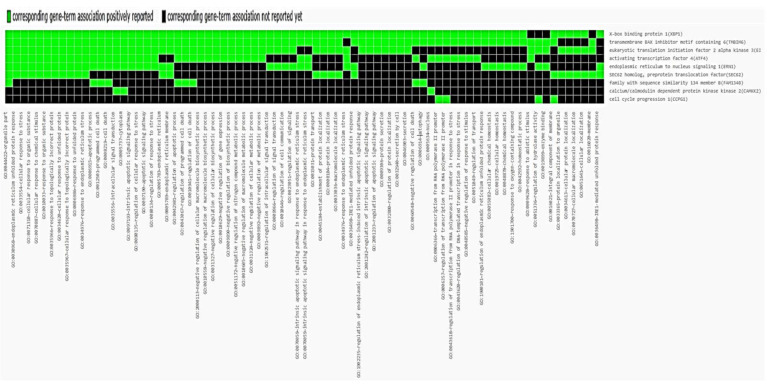
Analysis of clinical data using DAVID bioinformatics tools. Figure depicts the corresponding gene-term association of ER stress proteins, ERphagy proteins, and Ca^2+^ signaling proteins.

**Table 1 cells-11-03773-t001:** Apoptotic marker proteins and stress inducers in canonical and noncanonical ER stress responses in different cell lines.

Cell Death Mode	Apoptosis Marker	Stress Inducer	Canonical and Noncanonical Stress Response	Cell Type	Reference
Apoptosis	Mcl-1 (anti), Noxa	2-Deoxyglucose (2-DG)	ATF4	Rhabdomyosarcoma	[129]
Apoptosis	Puma, Noxa,	Tunicamycin, thapsigargin	IRE1α, ATF6 Protective effect	Melanoma cells	[130]
Apoptosis	Death receptor 5	Thapsigargin	PERK, ATF4-CHOP	MDA-MB231	[131]
Apoptosis	Noxa	Hypericin-based photodynamic therapy	PERK	T24 bladder carcinoma	[132]
Apoptosis	Noxa	Bortezomib	ATF4	Neuroectodermal tumor cells	[133]
Apoptosis	caspase-8 activation	Bortezomib/MG132		HEK293, MDAMB231, MCF7	[134]
Apoptosis	CHOP	Z36	p-PERK	HeLa cells	[135]
Apoptosis	Cleaved PARP	Brefeldin A (BFA)	C/EBP homologous protein (CHOP)	A549 cells	[136]
Apoptosis	Bax, Bak	Tunicamycin, Thapsigargin, Brefeldin A		MEF	[137]

**Table 2 cells-11-03773-t002:** FDA approved anticancer drugs that modulates ER stress-mediated autophagy.

Drug Name	Target	Types of Tumor	ER Stress	Autophagy	Reference
Bortezomib	Molecular chaperones Hsp70	Breast cancers	Integrated stress response (ISR) and UPR activation	eIF2α mediated induction of autophagy	[150]
Imiquimod (IMQ)	Toll-like receptor (TLR) 7 ligand	Basal cell carcinoma	PKR is activated to phosphorylate eIF2α	PKR markedly enhanced IMQ-induced conversion of LC3-I to LC3-II	[151]
Cucurbitacin B (CuB)	cell cycle in G_2_/M phases	Melanoma cells	Phosphorylation of eIF2α also mediates the conversion of LC3-I	CuB-induced autophagy was associated with c-Jun *N*-terminal kinase (JNK) activation	[152]
Sorafenib	Tyrosine kinase inhibitor	Hepatocellular carcinoma	PERK-ATF4 pathway correlation with drug resistance	Beclin 1 plays a role in ER stress-related autophagy	[153]
Oxaliplatin	DNA damage	Colon Cancer	Activation of UPR components	Enhanced autophagy genes (ATG5 or Beclin 1).	[154]
5-fluorouracil	Inhibition of thymidylate synthase (TS) and incorporation of its metabolites into RNA and DNA	Colon cancer	Activation of different signal branches in UPR	Protective autophagy is induced by Beclin-1 expression conversion of LC3I to LC3II.	[155,156,157]
Thapsigargin	Oxidative DNA damage	Osteosarcoma	PERK mediated cytoprotection	Inhibits mTORC1 activity and induces autophagy	[158]
Paclitaxel	ER stress-inducing agents against cancer	Breast cancer	IRE1α-ERK1/2 mediates the activation of RSK2	RSK2 enhanced autophagy	[159]
Sunitinib and Gemcitabine	Vascular endothelial growth factor receptor	Pancreatic ductal adenocarcinoma (PDAC)	GRP78 and XBP1 splicing mediated ER homeostasis.	Increased lysosomal enzymatic activity and autophagy	[106]
Brigatinib	Anaplastic lymphoma kinase (ALK) inhibitor	Non-small-cell lung cancer	IRE1α/JNK signaling	Enhanced ER-Phagy	[139]
Tunicamycin	Target calcium	Breast cancer	IRE1-TRAF2 complex formation	Autophagy regulated by IRE1/JNK/Beclin 1	[99]
Temozolomide (TMZ)	These agents act directly on DNA	Glioblastoma multiforme (GBM)	Activated PERK, XBP1	Autophagy genes (e.g., *ATG5*, *ATG7*, *BECN1*)	[160]
Verotoxin-1	ER stress inducer	Lymphoma cancer	ER stress response by IRE1 and ATF6—two ER stress sensors	Protective role through ER-phagy, depending on the cell line	[161]
Tetrahydrocannabinol (THC)	Stimulation of ER stress	Glioma cell	Eukaryotic translation initiation factor 2α (eIF2α) phosphorylation	ER stress response promotes autophagy	[162]
Cinnamomum cassia	ER stress inducer	Gastric cancer	ER stress-induced eIF2α/ATF4 axis induces AMP-activated protein kinase (AMPK) phosphorylation	Ca^2+^ release induced autophagy by Beclin 1, ATG5, and LC3B expression	[163]

**Table 3 cells-11-03773-t003:** Autophagy inhibitors intersecting with ER stress.

Name of Drug	Target	ER Stress	Class	Cancer Types	Reference
Chloroquine (CQ)	Fusion process of autophagosome and lysosome	Increase apoptosis via PERK-eIF2α-ATF4 pathway	Lysosomotropic agents	Pancreatic neuroendocrine neoplasms (PanNENs)	[228]
3 methyladenine (3-MA)	PI3K	Activated IRE1α-ER stress sensor	PI3K inhibitor	Colon & breast cancer	[99]
Hydroxychloroquine (HCQ)	Lysosomal cathepsin D	Not observed	Lysosome inhibitor	Gastric cancer	[229]
Bafilomycin A1	V-ATPase	ER stress via the IRE1 α -JNK pathway	Lysosomal H^+^-ATPase inhibitor	Gastric cancer cells	[230]
Elaiophylin	Inhibition of autophagy flux	Fetal ER stress-induced apoptosis	Autophagy flux inhibitor	Multiple myeloma (MM)	[231]
4-Acetylantroquinonol B	Inhibition of autophagy flux	Not observed	Autophagy flux inhibitors	Epithelial cancer cells	[232]
Thymoquinone	Permeabilization of the lysosome membrane	ER stress markers (GRP78, CHOP)	Lysosomotropic agents	Bladder cancer	[233]
S130	Target ATG4B	Not observed	Autophagy inhibitor	Colorectal cancer	[234]

**Table 4 cells-11-03773-t004:** Efficacy of ER stress inducer and autophagy inhibitors in cancer therapy.

Single Drug	Cancer Types	Dose	Mechanism	Combination	Target	Final Outcome	Ref
Vitexin	Breast cancer	Vitexin concentration 20 µM	Disrupts FAM134B-BiP complex inhibits ER-phagy and suppresses breast cancer progression	Tunicamycin	ER stress inducer	When used in combination with ER stress inducer, synergistically stunted cell proliferation	[103]
Temozolomide (TMZ)	Malignant Glioma	10–15 μmol/mL active concentration	calcium-mobilizing compound activated autophagy-related gene BECN1, ATG7	Chloroquine	Autophagy Inhibitor	CQ treatment-enhanced TMZ can synergize with the activation of Ca^2+^ signaling and reactive oxygen species (ROS)	[227]
Sunitinib	Ovarian Carcinoma	10 μm/mL sunitinib	Hypoxia–induce autophagy	Lys05 Active concentration 10 μmol/L	Autophagy Inhibitor	Autophagy inhibition as a strategy to overcome resistance to RTK inhibitors such as sunitinib	[235]
Brigatinib	Colorectal Cancer	2 μM Brigatinib	Activates ER-Phagy via ER stress-signaling pathway	Chloroquine	Autolysosome Inhibitor	Inhibition of ER-phagy enhances the susceptibility of CRC cells to brigatinib in vitro and in vivo	[139]
Carboplatin	Triple-negative breast cancer	24 mg/kg	DNA damage induces autophagy	Chloroquine	Autophagy Inhibitor	Autophagy inhibitor CQ causes sustained oxidative DNA damage	[196]
Cisplatin and Daunorubicin	Non-small cell lung cancer (NSCLC)	10 µM effective concentration	DNA alkylating agents	0.1 µM effective concentration of SBI0206965	Inhibition of Ulk1	Inhibition of Ulk1 suppresses NSCLC cell growth and sensitizes NSCLC cells to cisplatin by modulating both autophagy and apoptosis pathways	[236]
Fluorouracil (5FU)	Esophageal squamous cell carcinoma	5 mg/kg	Inhibits the nucleotide synthesis	LY294002 Effective concentration 2 µM	Autophagy Inhibitor	Autophagy inhibitor (LY) will disrupt the protective mechanism of cancer cells	[237]
Photodynamic therapy (PDT) aluminum phthalocyanine chloride	MEF cells	Active concentration 15 µM	photosensitizing agents target Ca^2+^ signaling	Not use		Generating oxidative stress capable of causing damage to cell membranes, proteins, or DNA. Increasing intracellular Ca^2+^concentration and activating the apoptotic pathway	[238]

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
