# Peer review of "Canonical and Noncanonical ER Stress-Mediated Autophagy Is a Bite the Bullet in View of Cancer Therapy"

_cells, 2022, doi:10.3390/cells11233773_

Round 1

Reviewer 1 Report

The role of autophagy, as catabolic process whose activation may help  cells  to adapt to cellular stress, has been well-recorded in various biological systems, especially in cancer.  However, in this review the authors have rightly pointed out,the interplay between autophagy and ER stress or UPR response.

This review well recapitulate the interplay between autophagy and ER stress or UPR response, the possible advantages or disadvantages of autophagy manipulation in cancer cells and  discusses possible target mechanisms for anticancer therapy. The manuscript is well written, the figures are clear and the reference list is extensive. However, it is my opinion that the article is too long and some of the paragraphs are too redundant, which makes this article much more like a chapter of a book than a review.  For istance the paragraphs entitled “Significance of molecular mechanisms regulating ER-phagy in selection of therapeutic targets” and “DNA damage response (DDR)” might be removed.

Author Response

We appreciate the important comment. We also felt that review article was a bit long. However, topics like canonical and non-canonical Er stress-mediated autophagy require a more in depth analysis and discussion to understand and address the UPR or non-canonical ER stress components to develop a better and more effective cancer treatment. Thus analysing and discussing the current understanding of these topics increased the article's lengths. Regarding cutting down some redundant descriptions, we believe that having these specific talks at that specific juncture is essential. If we cut them out, it might be hard to understand the relevant target-specific discussion. Hence, we would like to keep those descriptions in light of the significance of those redundant descriptions.

Reviewer 2 Report

1) All figures must be checked. For example, in figure 2 - calcineurin

2) If the balance of the intracellular concentration of Ca2+ ions is meant, then it should be used in the form [Ca2+]i

3) We need a figure in which in one part the canonical path of ER stress will be clearly indicated, and in the other part - non-canonical

Author Response

1) Thank you for identifying the spelling errors in the Figures. As suggested, we have corrected the misspelt words in all the figures.

2) As suggested, we have inserted [Ca2+] wherever it is applicable.

3) The review article is quite extensive and has the maximum number of figures to describe the mechanisms and data-supported explanations. Also we feel that adding another figure would add a technical load to the script. Moreover, the existing figures are adequate to address both canonical and non-canonical ER stress-mediated autophagy 

Round 2

Reviewer 1 Report

I feel that the article is still too long and that some paragraphs are too redundant, however I understand the authors' point of view and the difficulty of cutting out the redundant part without making the reader lose understanding of the specific objective of this review.

Author Response

I have attached the file here

Reviewer 2 Report

The article can be accepted for publication in its current form

Author Response

Attached file
